# Validity and reliability of the Balance Error Score System (BESS) Thai version in patients with chronic non-specific neck pain

Arisa Leungbootnak[1], Rungthip Puntumetakul[2,3], Thiwaphon Chatprem[2,3]*, Surachai Sae-Jung[4], Rose Boucaut[5]

1 Faculty of Associated Medical Science, Human Movement Sciences, School of Physical Therapy, Khon Kaen University, Khon Kaen, Thailand, 2 Faculty of Associated Medical Science, School of Physical Therapy, Khon Kaen University, Khon Kaen, Thailand, 3 Faculty of Associated Medical Science, Research Center in Back, Neck, Other Joint Pain and Human Performance (BNOJPH), Khon Kaen University, Khon Kaen, Thailand, 4 Faculty of Medicine, Department of Orthopedics, Khon Kaen University, Khon Kaen, Thailand, 5 UniSA Allied Health and Human Performance, University of South Australia, Adelaide, Australia

* thiwch@kku.ac.th

## Abstract

### Background

Neck pain has been found to affect the somatosensory system, which can lead to impaired balance control. To assess the balance of patients with neck pain and other conditions, the balance error scoring system (BESS) is commonly used as a static balance measurement tool. However, this tool is seldom used in Thailand due to its English language format.

### Objective

To translate and determine the content, convergent validity, and reliability of a Thai version of the BESS tool.

### Material and methods

A process of cross-cultural adaptation was utilized to translate BESS into a Thai version, called BESS-TH. To assess content validity, five physical therapy lecturers specializing in the musculoskeletal field used BESS to measure balance in participants with neck pain. For the convergent validity process, 130 patients diagnosed with chronic non-specific neck pain (CNSNP) were randomly assessed using four static balance tests (BESS, Single-leg balance test (SLBT), Romberg test, and Tandem stance test). For reliability, two assessors with varying years of work experience independently assessed videos of the participants twice using the BESS-TH, with a minimum 7-day interval between assessments.

### Results

The BESS-TH used to assess balance of patients with neck pain demonstrated acceptable content validity (index of item objective congruence (IOC) = 0.87). The Spearman's Rank Correlation Coefficient was calculated between the BESS-TH and three other measures:

**Funding:** This study received Research Funding for Supporting Lecturer to Admit High Potential Student to Study and Research on His Expert Program Year 2021 at Khon Kaen University. The funders had no role in study design, data analysis, decision to publish, or preparation of the manuscript. All authors involved in this study have contributed, read, and approved the manuscript.

**Competing interests:** The authors have declared that no competing interests exist.

the SLBT with eyes open and eyes closed, the Romberg test with eyes open and eyes closed, and the Tandem stance test with eyes open and Tandem stance test with eyes closed. The values obtained were as follows: -0.672, -0.712, -0.367, -0.529, -0.570, and -0.738, respectively. The inter-rater and intra-rater reliability were 0.922 (95% CI = 0.864–0.956) and 0.971 (95% CI = 0.950–0.983), respectively. Minimum detectable change (MDC) for the total BESS score of inter-rater and intra-rater reliability were 7.16 and 4.34 points, respectively.

## Conclusion

The BESS-Thai version was acceptable, reliable, and valid for evaluating balance performance in patients with CNSNP. This tool can be used and applied to clinically evaluate postural control in Thailand.

## Introduction

The term "neck pain" (NP) refers to the experience of pain and discomfort in the anatomical region of the neck, which may or may not be accompanied by pain in the head, trunk, and upper limbs [1]. Based on the Global Burden of Disease study from 1990 to 2019, NP is one of the top four musculoskeletal disorders in the world [2,3]. NP has a reported annual prevalence of 37.2% and a lifetime prevalence of 48.5%, which can lead to chronic neck pain (CNP) in all genders and ages [4] and when compared over the last two decades demonstrated increasing prevalence and incidence of NP [3]. In China [5], there is a high (86.3%) one-point prevalence of non-specific neck pain (NSNP), where NSNP is defined as a type of NP without a detectable etiology and with no features of red flag conditions such as: malignancy, infection, inflammation, myelopathy, other histories of orthopedics conditions and drop attack during head movement, or symptoms following whiplash [6,7]. In Thailand, NP has similarly been reported to have a high prevalence up to 81.9% and was one of the top three ranked musculoskeletal (MS) conditions in many occupations [8–13]. Further, NP is also associated with a significant financial burden for treatment within national healthcare systems in the United States and globally [14,15].

Neck problems in chronic non-specific neck pain (CNSNP) have the potential to significantly disrupt all aspects of an individual's physical, psychological, and social well-being [16]. Individuals experiencing neck pain may have disruptions in their sensory input [17–19], and abnormal muscle activity and muscle endurance in deep cervical muscles [20–22]. They may also suffer from alterations in the cervical structure, such as: fatty infiltration or muscle atrophy [23–26], modifications in head and eye movement regulation [17], limited range of motion [22], and compromised cervical kinesthesia [17,27]. These signs and symptoms manifest in the cervical spine, which features a highly intricate proprioceptive system that plays a pivotal role in the control of balance and correct posture [18]. Thus, individuals suffering from NP have diminished balance and changing walking patterns [17,28,29], leading to increase their susceptibility to falls and subsequent injuries [30–32]. Falling can pose a serious risk to patients, leading to fractures in the hip or lower extremities, head injuries, and fear of falling which can eventually limit daily activities. Less active individuals are more prone to falling and can develop weaker muscles, leading to longer hospital stays [32–37].

The concept of balance, or postural control, comprises the visual, vestibular, and somatosensory systems. These systems are seen as subsystems responsible for providing sensory

information to the central nervous system (CNS) [33]. Prior studies have indicated that individuals experiencing CNP exhibit reduced balance capabilities during static standing and dynamic walking tasks [34]. The presence of a balance disturbance can be attributed to a discrepancy between abnormal sensory input from the cervical region and normal sensory input from the visual and vestibular systems [17].

The selection of a suitable outcome measure for evaluating balance in individuals with NP poses a current challenge due to a lack of guideline recommendations [35]. Various research studies have utilized different tools in their research methodology to assess static balance. The tools applied in the previous research encompass force plates [28,36–38], Single Leg Balance Test (SLBT) [39], Romberg test [36], Tandem stance test [40,41], and balance error scoring system (BESS) [42].

The BESS tool was developed by researchers at the University of North Carolina for clinicians to evaluate postural stability [43]. It has been utilized in various studies to examine balance in different populations. Specifically, BESS has been employed in investigations involving: athletes [44,45], athletes with a sports-related concussion [46–48], individuals with ankle injuries [49–51], healthy participants [52,53], community-dwelling adults [54–56], and individuals experiencing NP [42]. The BESS balance measurement has a variety of subtests. The level of difficulty is heightened, and the task is intensified through the reduction of support and alteration of the standing surface. Wah and co-workers (2021) employed the BESS to assess balance in patients with NP [42]. The researchers justified their choice of static balance tool by highlighting the clinical applicability, simplicity, affordability, and practicality of BESS for evaluating postural stability [43].

Previous research using BESS has been conducted exclusively in English-speaking populations with NP. However, there have been no studies to date on the validity and reliability of the BESS when translated into Thai for individuals with neck pain.

To implement the BESS test among Thai individuals with chronic non-specific neck pain (CNSNP), it is crucial to first translate the BESS test into the Thai language. Following this, the content of the test should be assessed for its validity and reliability among individuals with CNSNP. It is also necessary to evaluate the convergent validity of BESS with other balance tests, such as the SLBT, the Romberg Test, and the Tandem Stance Test. These tests will serve as comparators to determine the effectiveness and accuracy of the BESS test.

## Materials and methods

The study was conducted from April 2023 to September 2023 with approval from the Local Centre for Ethics in Human Research (HE652087) of Khon Kaen University, Thailand. In addition, the study was also registered in Thai Clinical Trial (TCTR20230405003). Prior to the translation procedure, the investigators obtained authorization from the original BESS developers through email. Participants were required to sign an informed consent form before participating in the study, which was divided into four phases: 1) translation and development of the BESS Thai version (BESS-TH); 2) testing content validity; 3) testing convergent validity; and 4) testing the reliability of BESS-TH in patients with CNSNP.

### Participants

The study recruited participants through direct contact with the translator, expert physiotherapist, and advertisements such as posters and social media for physical therapists and CNSNP patients. Both men and women were eligible to participate. The study consisted of four phases, with each phase having its own group of participants. The breakdown of participants for each phase is as follows:

**Phase I: Translation procedure.** In the first phase of translation and cross-cultural adaptation, the researchers followed the guidelines from Beaton and coworkers (2010) and Sousa and Rojjanasrirat (2011) [57,58], which required the first group of participants, who are five bilingual native translators (Thai and English) with or without medical background. The second group of participants required 30 physical therapists, who are Thai native speakers, along with additional participants who performed a psychometric evaluation of the preliminary version of the translated tool on physical therapists who represent the target population [58].

**Phase II: Content validity for assessing BESS-TH in people with chronic non-specific neck pain.** The third group of participants comprised five experienced physical therapists with more than 10 years of experience who measured the content validity of the BESS-TH for use with CNSNP participants.

**Phase III: Convergent validity of BESS-TH in people with chronic non-specific neck pain.** The fourth group of participants were participants with CNSNP who lacked an identifiable cause and did not exhibit any symptoms of serious underlying illnesses. They had to meet the following inclusion requirements: 1) neck pain duration for at least three months; 2) aged between 20 and 69 years; 3) body mass index (BMI) of below thirty kg/m$^2$; 4) mild to moderate pain on the visual analog scale (VAS; 5–74 mm); and 5) excellent cooperation and communication in the Thai language. Participants were excluded if they had any of the following: a history of visual, auditory, vestibular, or neurological deficits, head or neck injuries, cervical or thoracic spinal surgery caused by trauma or lower limb surgery within the past year, or severe neurological or psychiatric disease, chronic lower-extremity musculoskeletal disorders, fractures, and injuries, medical conditions that could adversely affect balance performance, or alcohol or sedative drug use within the prior 48 hours [34,42,55,56,59]. For convergent validity measurement, the correlation formula ($n = \left(\frac{Z_{\alpha/2}+Z_\beta}{Z_{(r)}}\right)^2 + 3$; $Z_{(r)} = \frac{1}{2}\ln\left(\frac{1+r}{1-r}\right)$) was used to calculate the sample size [60]. This study sets the α value at 0.05, the β value at 0.1, and the value r at 0.31 [43], so the participant number required for the validity study was 105, and considering the 10% drop-out rate, the total number of participants was 130. Convergent validity was determined when the participants performed the BESS-TH test components, and the other three static balance tests were conducted with 130 Thai participants with CNSNP.

**Phase IV: Reliability testing.** For reliability measurement, we followed the epidemiology paper for calculating the sample size based on the ICC estimation. This study set the 95% confidence interval width (CIW) at 0.2, the number of measurements per individual (k) at 2, the alpha value at 0.5, and the estimated ICC at 0.8. Thus, the number of reliabilities was measured in 51 CNSNP participants [61].

## Procedure

The procedure included four consecutive phases: (1) translating and developing the BESS Thai version (BESS-TH), (2) assessing content validity, (3) convergent validity, and (4) reliability testing in CNSNP.

**Phase I: Translation and development of the Thai version of the BESS (BESS-TH).** BESS-TH was translated and cross-culturally adapted in accordance with standard guidelines [57,58]. The translation guideline includes six steps, as follows:

1. **Forward translation:** Three Thai-native bilingual translators, including two physiotherapists with years of experience and one specialist English translator with no medical or physical therapy background, translated the original material from English to the Thai versions of BESS (THAI-1, THAI-2, and THAI-3).

2. **Synthesis I:** Five people (two translators of the first step and three members of the research committee) combined all three versions of step 1 (THAI-1, THAI-2, and THAI-3) with the original version by comparing and contrasting each of their translations and coming to an agreement on wording to clear up word ambiguity. This allowed them to create the next Thai version of the test (THAI-123).

3. **Backward translation:** This was conducted by two English-native bilingual translators who could read Thai and had never seen the original versions of the test's construction. In the process, two back-translated versions of the BESS (BT1 and BT2) were developed by the researchers.

4. **Synthesis II, or comparative analysis of the two back-translated versions and the original version:** The same expert committee evaluated all versions of the BESS, including the original version, THAI-123, BT1, and BT2, and formulated the preliminary BESS-TH (prefinal BESS-TH).

5. **Testing of the Prefinal Version:** This process was tested with individuals fluent in the instrument's Thai language to assess the clarity of instructions, response format, and items. This process required 30 physical therapists to use the prefinal version of the BESS-TH test during the evaluation of balance performance in patients with NP to determine how well the test's instructions or items were clear and readily comprehended in a clinical setting.

6. **Final version of BESS-TH:** The same expert committee evaluated all previous feedback. The expert committee amended an item if 20% or more of the items that participants mentioned misinterpreted the language [58]. The researchers then modified BESS-TH and tested its comprehensibility. They finalized the BESS-TH.

**Phase II: Content validity for assessing BESS-TH in people with chronic non-specific neck pain.**  The index of item objective congruence, also known as the IOC, provided content validity. A panel consisting of five academic and experienced physical therapists examined the test's content validity. The six subtests of the BESS balance test were evaluated, and the score determined by the BESS-TH test to measure balance in participants with CNSNP was rated. The experts were able to rank the relevance of each subtest on an ordinal scale (+1, 0, or -1) according to the level to which it was related to the objectives of the study. The rating score of the expert evaluation was as follows: consistent with the study's objective +1, non-consistent with the study's objective -1, and unclear 0 [62]. The IOC of each item was determined by dividing the total score by five and then multiplying the result by one hundred. If the value of the IOC ranges between 0.5 and 1, it indicates that the subtest was either measured on purpose or that it is applicable to that objective [63].

**Phase III: Convergent validity of BESS-TH in people with chronic non-specific neck pain.**  During the measurement of the same concept with a different test or variable, the correlation values should be found in the same direction. The static balance tests that were used to measure patients with NP included BESS, SLBT, Romberg test, and Tandem stance test [36,39–42]. The Pearson rank correlations, or Spearman rank correlations, were used to consider the correlation coefficient between the tests depending on normal or non-normal data distribution respectively. The values of the correlation coefficient can be classified into very high correlation (0.91–1.00), high correlation (0.71–0.90), normal correlation (0.51–0.70), low correlation (0.30–0.50), and very low correlation (0.00–0.30) [64]. The participants were given a 5-minute break between tests, and the results were recorded.

**Phase IV: Reliability testing.**  Research assistants recorded a video clip of the BESS test administered to individuals with CNSNP. Participants with CNSNP had a single video recording taken of them. Researcher 1 (5 years of experience) and Researcher 2 (30 years of experience) evaluated inter-rater reliability using the BESS score from a video recording. According to intra-rater reliability testing, Researcher 1 randomized the order of watching the clips and

re-evaluated the intra-rater reliability of BESS measurement at least 7 days after the initial video viewing [65–68]. The ICC interpretation can be represented as follows: ICC values below 0.5 indicate poor reliability; values between 0.5 and 0.75 represent moderate reliability; values between 0.75 and 0.9 represent good reliability; and values above 0.90 represent excellent reliability [69]. The reliability process can calculate the standard error of measurement (SEM) representing the random variation of an individual's scores over repeated assessments [70], and Minimum detectable change (MDC) for referring to the amount of variable change needed to be confident that the error did not cause the entire observed difference and that some real change happened [71].

## Outcome measures

The static balance tests used to measure balance in participants with neck pain included the BESS, SLBT, Romberg test, and Tandem stance tests. The detail of each test follows:

**Balance error score system Thai version (BESS-TH).** BESS was developed to evaluate postural stability without using complex or expensive equipment [43]. BESS is a brief, and easily administered static balance test [54]. The BESS consists of 3 stances: a double-leg stance (hands on the hips and feet together), a single-leg stance (standing on the nondominant leg with hands on hips), and a Tandem stance (nondominant foot behind the dominant foot) in a heel-to-toe fashion (Fig 1).

The stances are performed on a firm surface and on a foam surface with the eyes closed, with errors counted during each 20-second trial. An error is defined as opening eyes, lifting hands off hips, stepping, stumbling, or falling out of position, lifting the forefoot or heel, abducting the hip by more than 30˚, or failing to return to the test position in more than 5 seconds. Each subtest has a maximum of 10 scores. The total summation score can range from 0 (no error) to 60 (severe static balance) [43,72].

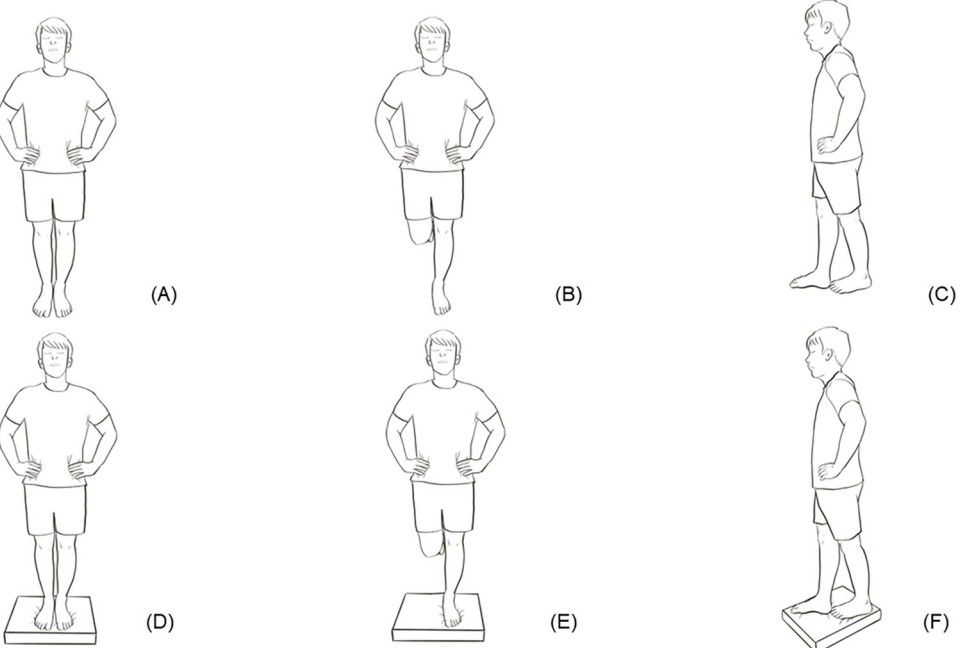

**Fig 1.** Stances used in BESS: Double-leg stance on firm surface (A); single-leg stance on firm surface (B); Tandem stance on firm surface (C); double-leg stance on foam surface (D); single-leg stance on foam surface (E); Tandem stance on foam surface (F).

**Single-leg balance test (SLBT).**    For an SLBT, participants stand on one foot with the knee of the other leg bent and not contacting the opposite leg, as displayed in Fig 2. The participant is placed in a testing position and told to stay balanced for 45 seconds. During testing, if the participant's raised leg touches the limb being tested or has movements such as jumping on one leg or touching something to assist with balance, he or she will be disqualified, and the researcher will immediately stop testing. The test is performed with eyes open and closed, and the 3-trial times of the test were written down and evaluated to determine the mean value [39,73]. This test demonstrates moderate reliability (ICC = 0.60–0.81) [74] and good reliability (ICC = 0.898) [75].

**Romberg test.**    Participants conducted the Romberg test by standing with their feet together (toes and heels nearby together) and their hands crossed at the chest, as shown in Fig 3. Standing with eyes open and eyes closed for 30 seconds, three times, and averaging the results. This test has demonstrated good reliability (ICC = 0.86) [76].

**Tandem stance test.**    The participant performs the Tandem stance test by standing with their dominant foot behind their non-dominant foot on a firm surface, as shown in Fig 4. The participants could stand with their eyes open and closed and the researcher noted the time that could be performed (in seconds). If they were unable to maintain a stable stance for thirty seconds, this would indicate an anomalous balance performance [77]. This test has been reported to have good reliability (ICC = 0.86) [76].

## Data analysis

Statistical Package for Social Sciences (SPSS) (version 28; IBM Corp., Armonk, NY) was used to analyze the data. Descriptive statistics were used to report the demographic characteristics. The normal distribution was tested using the Kolmogorov-Smirnov test. Content and convergent

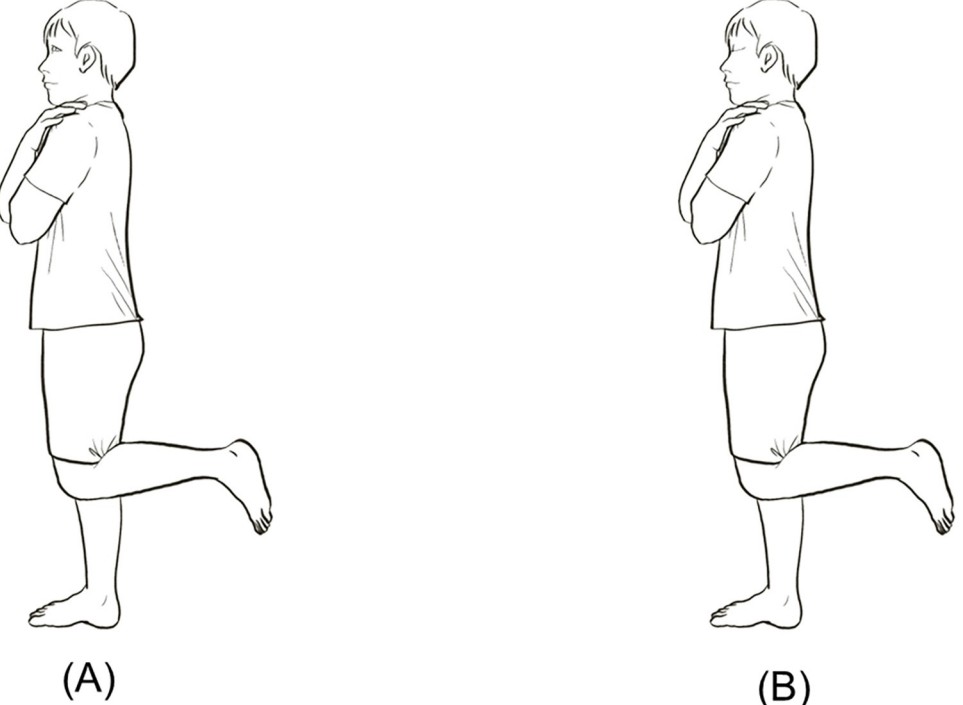

**Fig 2.**  Stances used in SLBT: Eyes open (A); eyes closed (B).

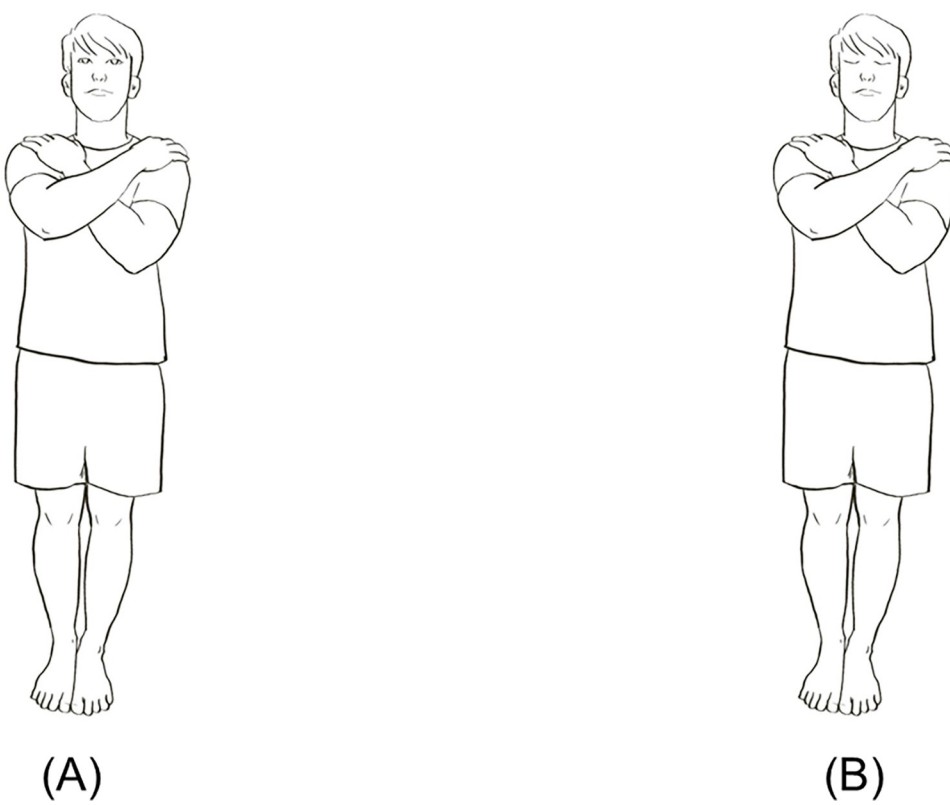

**Fig 3.** Stances used in the Romberg test: Eyes open (A); eyes closed (B).

validity were measured using IOC and Spearman rank correlations, respectively. Inter-rater and intra-rater reliability were measured using the intraclass correlation coefficient (ICC) model 2,1 and the ICC model 3,1. SEM was calculated to measure the data variation over reassessment and MDC was used to measure the amount of confident variable reel change.

## Results

### Demographic characteristics

**Physical therapy in the cross-cultural translation phase.** Demographic information of the 30 physical therapists who took part in the cross-cultural procedure is presented in Table 1. The average age of the participants was 27.6 ± 3.2 years, with females comprising 73.3% and males comprising 26.7% of the participants in the study. The average term of working in the field of physical therapy was 57.1 ± 35.3 months.

**Neck pain participants in the validity and reliability phase.** The validity sample comprised 130 patients with CNSNP. The demographic information and pain history details are shown in Table 2. The average age of the participants was 44.41 ± 14.25 years. Of the participants, 66.2% were female and 33.8% were male. The average duration and intensity of the pain were 52.62 ± 52.05 months and 4.37 ± 1.83 scores, respectively.

### Content validity

For all 6 items, content validity for the BESS test reached an average IOC of 0.87 (range 0.60–1.00), as shown in Table 3. This value showed acceptable validity [63].

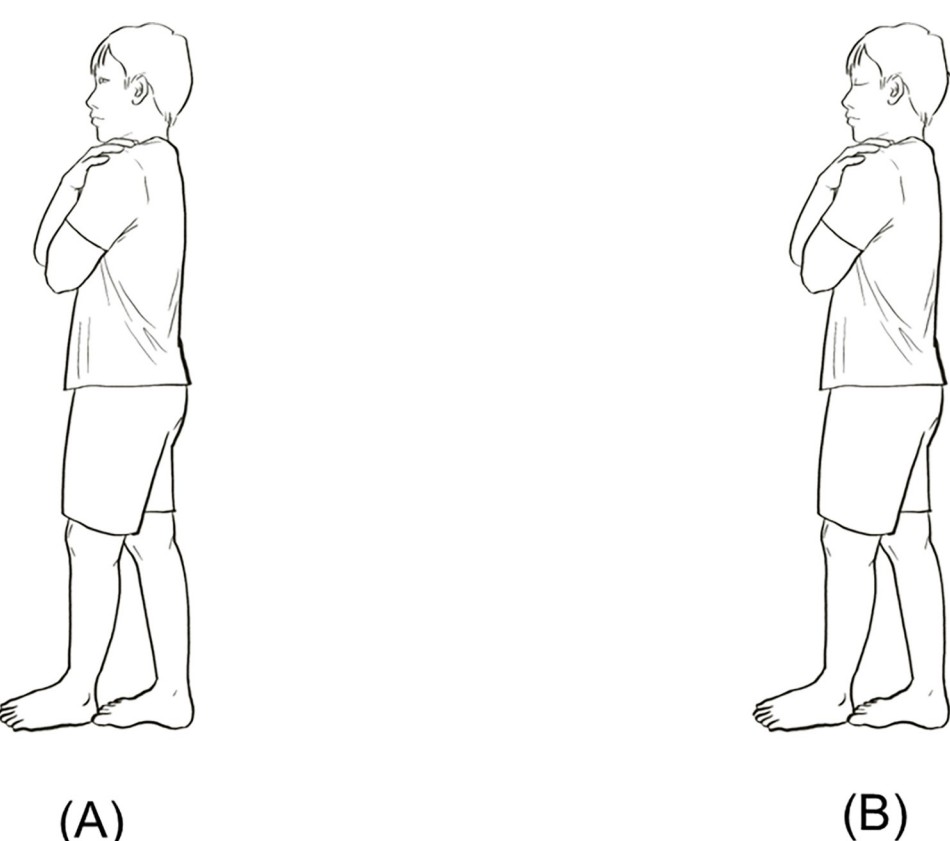

**Fig 4.** Stances used in Tandem stance: Eyes open (A); eyes closed (B).

### The score of each measurement

The scores of the measurements, including the mean, standard deviation, minimum score, and maximum scores for the BESS, SLBT, Romberg, and Tandem stance tests, are shown in Table 4. The average values of BESS-TH, SLBT with eyes open, SLBT with eyes closed, Romberg with eyes open, Romberg with eyes closed, Tandem stance with eyes open, and Tandem stance with eyes closed were 22.36 ± 8.61, 30.0 ± 12.6, 11.4 ± 8.7, 29.7 ± 1.2, 24.9 ± 7.5, 27.5 ± 4.8, and 17.9 ± 8.7, respectively.

### Convergent validity assessment

For convergent validity, the BESS-TH exhibited a high negative correlation (r = -0.712) with SLBT with eyes closed and Tandem with eyes closed (r = -0.738), moderate correlation (r =

**Table 1. Demographic characteristics of the physical therapists (n = 30) for cross-cultural adaptation process.**

| Demographic Characteristics | n (%) | Mean ± SD | Range |
|---|---|---|---|
| Gender<br>Male<br>Female | <br>8 (26.7)<br>22 (73.3) | | |
| **Age (years)** | | 27.6 ± 3.2 | 23–36 |
| **Education level**<br>**Bachelor's degree**<br>**Master's degree**<br>**Doctoral degree** | <br>14 (46.7)<br>13 (43.3)<br>3 (10.0) | | |
| **Working experience (months)** | | 57.1 ± 35.3 | 4–129 |

**Table 2. Demographic characteristics of the 130 participants for the convergent validity process.**

| Demographic Characteristics | n (%) | Mean ± SD | Range |
|---|---|---|---|
| Gender<br>Male<br>Female | <br>44 (33.8)<br>86 (66.2) | | |
| Age (years) | | 44.4 ± 14.3 | 20–69 |
| BMI (kg/m²) | | 23.8 ± 3.4 | 18.59–29.97 |
| Education<br>No<br>Primary school<br>High school<br>University | <br>2 (1.5)<br>37 (28.5)<br>43 (33.1)<br>48 (36.9) | | |
| Pain duration (months)<br>3 months– 1 years<br>> 1 years | <br>25 (19.2)<br>105 (80.8) | 52.6 ± 52.1 | 3–384 |
| Referred pain<br>No<br>Yes | <br>117 (90)<br>13 (10) | | |
| VAS | | 4.37 ± 1.83 | 0.52–7.40 |

BMI: Body Mass Index.

VAS: Visual Analogue Scale.

-0.51 to -0.70) with SLBT with eyes open, Romberg with eyes closed and Tandem with eyes open, and low correlation (r = -0.31 to -0.50) with Romberg with eyes opened (Table 5). Further, there were statistically significant associations (p<0.001) observed between the BESS-TH and the other balance outcome measurements.

## Rater reliability consideration

Table 6, the inter-rater reliability of this BESS-TH demonstrated a high level of agreement, with a calculated value of 0.922 (95% CI = 0.864–0.956). Regarding intra-rater reliability, the inexperienced physical therapist employed a tool twice to assess the video clip, ensuring a 7-day interval between measurements to mitigate any potential recollection of prior measurements. The results demonstrated excellent intra-rater reliability, as indicated by a high agreement value of 0.971 (95% CI = 0.950–0.983). The MDC of inter-rater and intra-rater reliability were 7.16 and 4.34 points, respectively.

**Table 3. The content validity of each item of the BESS test for chronic non-specific neck pain patients (expert committee review).**

| Items | Detail | IOC |
|---|---|---|
| 1 | Double-leg stance on a firm surface | 0.6 |
| 2 | Single-leg stance (standing on the non-dominant leg) on a firm surface | 0.8 |
| 3 | Tandem stance (non-dominant leg behind dominant leg) on a firm surface | 1.0 |
| 4 | Double-leg stance on a foam surface | 1.0 |
| 5 | Single-leg stance (standing on the non-dominant leg) on a foam surface | 0.8 |
| 6 | Tandem stance (non-dominant leg behind dominant leg) on foam surface | 1.0 |
| | Average | 0.87 |

IOC: Index of item-objective congruence.

**Table 4. The score of each balance measurement.**

| Measurements | Mean ± SD | Min | n (%) | Max | n (%) |
|---|---|---|---|---|---|
| **BESS-TH (scores)** | 22.36 ± 8.61 | 4 | 1 (0.8) | 46 | 1 (0.8) |
| **SLBT with eyes open (seconds)** | 30.0 ± 12.6 | 3.85 | 1 (0.8) | 45 | 29 (22.3) |
| **SLBT with eyes closed (seconds)** | 11.4 ± 8.7 | 1.31 | 1 (0.8) | 45 | 1 (0.8) |
| **Romberg with eyes open (seconds)** | 29.7 ± 1.2 | 21 | 1 (0.8) | 30 | 120 (92.3) |
| **Romberg, eyes closed (seconds)** | 24.9 ± 7.5 | 4 | 2 (1.5) | 30 | 79 (60.8) |
| **Tandem with eyes open (seconds)** | 27.5 ± 4.8 | 4.82 | 1 (0.8) | 30 | 84 (64.6) |
| **Tandem with eyes closed (seconds)** | 17.9 ± 8.7 | 1.5 | 1 (0.8) | 30 | 34 (26.2) |

BESS-TH: Balance error score system Thai version.

SLBT: Single-leg balance test.

## Discussion

The BESS test is used to measure static balance in people with various occupations or conditions. However, there is currently no valid and reliable Thai version of this test. This study shows that the BESS-TH has acceptable content validity (IOC = 0.87) and high correlation with SLBT with eyes closed (r = -0.712) and Tandem with eyes closed (r = -0.738). It also has a moderate correlation (r = -0.51 to -0.70) with SLBT with eyes open, Romberg with eyes closed, and Tandem with eyes open, and a low correlation (r = -0.31 to -0.50) with Romberg with eyes open. This study also found that the balance error score system Thai version (BESS-TH) has excellent inter-rater and intra-rater reliability (ICC of inter-rater reliability = 0.922, intra-rater reliability = 0.971), and reported MDC of inter-rater reliability at 7.16 points, and intra-rater reliability at 4.34 points among participants with CNSNP.

The IOC of content validity of BESS-TH in this study demonstrated IOC value in the range from 0.60 to 1.00 and the average IOC score of BESS-TH was 0.87. Each item possessing an IOC index greater than or equal to 0.5, was considered acceptable [63]. The average IOC greater than 0.75, indicates good content validity [62]. This means that each subscale of the BESS-TH is appropriate for use as an indication of balance ability.

The convergent validity of this study demonstrated a low to high negative correlation with the other 3 balance tests (-0.367 to -0.738). The result of this study demonstrated a similar correlation when compared with a gold standard called "force plate" represented in target sway (r = 0.31 to 0.79) [43], which represents a low to high correlation [64]. When examining the correlation between tests having the eyes open and closed, the tests conducted with closed eyes exhibited a higher correlation. One possible explanation for this difference may be explained by the fact that the BESS-TH was conducted with participants' eyes closed, in accordance with the instructions provided in the BESS test [43]. The participants of the BESS-TH test were given a slightly different protocol on their tested leg whilst performing the SLBT and Tandem

**Table 5. Correlation between BESS and other balance tests (n = 130).**

| Correlation | BESS | SLBT with eyes open | SLBT with eyes closed | Romberg with eyes open | Romberg with eyes closed | Tandem with eyes open | Tandem with eyes closed |
|---|---|---|---|---|---|---|---|
| **BESS** | 1.00 | -0.627* | -0.712* | -0.367* | -0.529* | -0.570* | -0.738* |

*Statistically significant, p < 0.01.

BESS: Balance error score system.

SLBT: Single-leg balance test.

**Table 6. Reliability of BESS-TH.**

| Item | N | Mean ± SD | | ICC | 95%CI | SEM | MDC |
|---|---|---|---|---|---|---|---|
| | | Rater1 | Rater2 | | | | |
| **Inter-rater reliability** | 51 | 18.86 ± 6.36 | 19.16 ± 6.72 | 0.922 | 0.864–0.956 | 1.83 | 7.16 |
| BESS firm surface | 51 | 4.37 ± 3.46 | 4.73 ± 3.76 | 0.899 | 0.825–0.943 | 1.15 | 4.50 |
| BESS foam surface | 51 | 14.49 ± 4.60 | 14.43 ± 4.71 | 0.915 | 0.851–0.952 | 1.36 | 5.32 |
| **Intra-rater reliability** | 51 | 19.02 ± 6.63 | 18.86 ± 6.36 | 0.971 | 0.950–0.983 | 1.11 | 4.34 |
| BESS firm surface | 51 | 4.49 ± 3.46 | 4.37 ± 3.46 | 0.969 | 0.946–0.982 | 0.61 | 2.39 |
| BESS foam surface | 51 | 14.53 ± 4.46 | 14.49 ± 4.60 | 0.952 | 0.918–0.972 | 0.99 | 3.89 |

SEM: Standard error of measurement.

MDC: Minimum detectable change.

subtests. This is a worthwhile observation that may have affected the outcome. This is another reason why the correlation between BESS-TH and other tests may not be shown to be highly significant. The present study observed a notably weak correlation with the Romberg test with the eyes open. This can be related to the presence of ceiling effects in this particular balance test, as evidenced by the maximum time of 30 seconds achieved by 92.3% of the 130 participants shown in Table 7. Therefore, when participants were then asked to perform this test with their eyes closed, the 30-second benchmark was implemented with possible expectations that they would not be able to balance for longer than the 30-second set time. It is noteworthy that among healthy participants, the Romberg test was the only assessment that did not exhibit a correlation with the force plate, even when performed with the eyes closed as part of the BESS test. This lack of correlation can be attributed to the absence of errors in the Romberg test among healthy individuals [43].

The intra-rater reliability of the present study displayed comparable levels of reliability to a study conducted by Wah and colleagues (2021), which assessed in participants with neck pain (inter-rater reliability = 0.98–0.99 and intra-rater reliability = 0.97–0.99) [42]. Additionally, the intra-rater reliability of the current study was found to be similar to that observed in other populations, including young adults (ICC = 0.92) [67], healthy youth athletes (ICC = 0.87–0.98) [65], and children (ICC = 0.96) [68]. This study had a higher intra-rater reliability score than that conducted on athletes by Finnoff and colleagues (2009), which showed moderate reliability (ICC = 0.74) [66], and college students conducted by Susco (2004) (ICC = 0.63–0.82) [52]. Several studies have assessed the reliability of measurements through the utilization of live comparisons of recorded videos [52,65,67] and the measurement of video recordings [66,68]. Utilizing video records to assess intra-rater reliability has the potential to enhance the total reliability of the measurements. In the same way, the inter-rater reliability between two

**Table 7. Ceiling and floor effect statistics for each balance measurement.**

| Measurements | Floor effect (%) | Ceiling effect (%) |
|---|---|---|
| BESS-TH (scores) | 0 | 0 |
| SLBT with eyes open (seconds) | 0 | **22.3** |
| SLBT with eyes closed (seconds) | 0 | 0.8 |
| Romberg with eyes open (seconds) | 0 | **92.3** |
| Romberg with eyes closed (seconds) | 0 | **60.8** |
| Tandem with eyes open (seconds) | 0 | **64.6** |
| Tandem with eyes closed (seconds) | 0 | **26.2** |

raters was demonstrated by watching the randomly sequenced video recording, which demonstrated good reliability. The inter-rater reliability demonstrates a similar value with previous studies performed to measure BESS in athletes (ICC = 0.78–0.96) [43], and greater value than previous research studies performed in athletes (ICC = 0.57) [66], university students (ICC = 0.66) [78], healthy children (ICC = 0.93) [68] and concussion patients (ICC = 0.80) [79]. A reason to support the lower interrater reliability might be because the rater may have a difference in years of experience and experience in different fields. The MDC of this current study demonstrated a lower score compared with the previous reliability study conducted by Finnoff and coworkers (2009) who reported an MDC of 9.4 (inter-rater) and 7.3 (intra-rater) points. This is because the MDC is based on SEM and ICC values [71].

This study is still subject to limitations about the feasibility of conducting the test in cases of mild to moderate severity of pain level. Therefore, it is not possible to generalize the findings of this study to persons experiencing severe pain. The process of translating BESS into the Thai language during cross-cultural adaptation holds significant implications for clinicians and physical therapists. Therefore, when considering the application of personal usage, it is necessary to assess the validity of utilization in different populations. This study included participants with chronic non-specific neck pain (CNSNP) who met the criteria of neck pain without an identifiable cause and had no indications of serious medical conditions such as cancer, infection, inflammation, myelopathy, previous orthopedic conditions, drop attack during head movement, or symptoms following whiplash injury. Our focus was solely on medical histories, meaning there may have an opportunity for a person with mild cervical disc herniation to have been included in the study. In future studies, investigators may consider the possibility of employing imaging techniques to rule out disc herniation. Additionally, this study did not include a comparison with a gold standard; hence, future research should incorporate measures of criterion-related validity.

## Conclusion

Based on the results of this study, it can be concluded that the Thai version of the Balance Error Score System (BESS-TH) is a reliable and valid tool for assessing static balance in individuals with chronic non-specific neck pain. The BESS-TH exhibits acceptable levels of content validity, convergent validity, and reliability, and can be confidently used by clinicians as one of their static balance assessment tools for patients experiencing mild to moderate levels of neck pain.

## Supporting information

**S1 Data. Analysis data.**
(XLSX)

**S1 Text. A Thai version of the balance error scoring system.**
(PDF)

## Acknowledgments

The researchers are thankful to the owner of the original version of BESS who allowed us to translate it into Thai, all experienced native translators who performed in the translation process, and experienced physical therapists in the content validity process. Additionally, this endeavor would not have been possible without the generous support from the participants who were willing to participate in this study.

## Author Contributions

**Conceptualization:** Arisa Leungbootnak, Rungthip Puntumetakul, Thiwaphon Chatprem.

**Data curation:** Arisa Leungbootnak.

**Formal analysis:** Rungthip Puntumetakul.

**Funding acquisition:** Rungthip Puntumetakul.

**Investigation:** Arisa Leungbootnak, Thiwaphon Chatprem.

**Methodology:** Arisa Leungbootnak.

**Project administration:** Arisa Leungbootnak, Rungthip Puntumetakul.

**Supervision:** Rungthip Puntumetakul, Thiwaphon Chatprem.

**Visualization:** Surachai Sae-Jung.

**Writing – original draft:** Arisa Leungbootnak.

**Writing – review & editing:** Rungthip Puntumetakul, Thiwaphon Chatprem, Surachai Sae-Jung, Rose Boucaut.

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
