## [Decision Letter · Decision Letter 0]

16 Jan 2024

PONE-D-23-41694Validity and reliability of the Balance Error Score System (BESS) Thai version in patients with chronic non-specific neck painPLOS ONE

Dear Dr. Chatprem,

Thank you for submitting your manuscript to PLOS ONE. After careful consideration, we feel that it has merit but does not fully meet PLOS ONE’s publication criteria as it currently stands. Therefore, we invite you to submit a revised version of the manuscript that addresses the points raised during the review process.

We look forward to receiving your revised manuscript.

Kind regards,

Mehrnaz Kajbafvala, Ph.D

Academic Editor

PLOS ONE

Journal Requirements:

2. Thank you for stating the following financial disclosure: "Research Fund for Supporting Lecturer to Admit High Potential Student to Study and Research on His Expert Program Year 2021".

3. In the online submission form, you indicated that "Data cannot be shared publicly because the authors want to conceal participants' information. However, if there are requirements for our data for academic proposals, they are available from the request at the corresponding email."

Reviewers' comments:

Reviewer's Responses to Questions

**Comments to the Author**

1. Is the manuscript technically sound, and do the data support the conclusions?

Reviewer #1: Yes

Reviewer #2: Partly

2. Has the statistical analysis been performed appropriately and rigorously? 

Reviewer #1: Yes

Reviewer #2: I Don't Know

3. Have the authors made all data underlying the findings in their manuscript fully available?

Reviewer #1: Yes

Reviewer #2: Yes

4. Is the manuscript presented in an intelligible fashion and written in standard English?

Reviewer #1: Yes

Reviewer #2: Yes

5. Review Comments to the Author

Reviewer #1: Thank you for the opportunity to review this manuscript "Validity and reliability of the Balance Error Score System (BESS) Thai version in patients with chronic non-specific neck pain". The manuscript was well written and is appropriate for publication in PLOS ONE Journal.

Reviewer #2: Dear Authors

Thanks for your interesting article. It is very useful in clinical work as well as research.

I have some suggestions as follows:

1. How did you diagnose the patients to have non-specific neck pain? I mean how did you assure that these patients did not have disc herniation or so on?

2. Please mention about the “Floor or ceiling effects” in the article. It should be calculated.

3. In addition to ICC, which has reported in the manuscript, please add the Cronbach’s alpha as well as minimally metrically detectable change (MMDC) for reliability and internal consistency.

4. Please check the spell of the word “Kolmogorov–Smirnov test”.

5. As you mentioned, you have not checked the criterion validity or using a gold standard to check the score. So, it seems that this article is somehow a cross-cultural adaptation. So, why did not mention in the topic?

6. What about the “Dimensionality”? Have you checked it?

6. PLOS authors have the option to publish the peer review history of their article (what does this mean?). If published, this will include your full peer review and any attached files.

Reviewer #1: No

Reviewer #2: No

---

## [Author Response · Author response to Decision Letter 0]

14 Feb 2024

Response to Reviewers 

Response: Thank you for providing this link, we have read and edited our manuscript to match the PLOS One format requirements.

2. Thank you for stating the following financial disclosure: "Research Fund for Supporting Lecturer to Admit High Potential Student to Study and Research on His Expert Program Year 2021".

Response: Thank you for your response and suggestion. We amended this sentence in the Cover Letter file as follows.

“This study received Research Funding for Supporting Lecturer to Admit High Potential Student to Study and Research on His Expert Program Year 2021 at Khon Kaen University. The funders had no role in study design, data analysis, decision to publish, or preparation of the manuscript.”

Section: Cover letter

3. In the online submission form, you indicated that "Data cannot be shared publicly because the authors want to conceal participants' information. However, if there are requirements for our data for academic proposals, they are available from the request at the corresponding email."

Response: Firstly, the authors apologize for our misunderstanding of your guideline and replying “No data sharing.” Forthwith, the authors are freely available to share the supporting file and have added the heading “Supporting information” in the manuscript. 

Section: Supporting information; Page 22; Line 460-464

Response: Thank you for reminding us of this point. We have carefully rechecked our reference lists and edited some references for the subsequence item to fit with the “Vancouver” style as detailed in the Reference style part of the PLOS ONE submission guideline. There was one reference that had been redacted and needed to be replaced. That is citation number 76 written as “Khasnis A, Gokula RM. Romberg’s test. J Postgrad Med. 2003;49: 169. Available: https://www.jpgmonline.com/article.asp?issn=0022-3859;year=2003;volume=49;issue=2;spage=169;epage=72;aulast=Khasnis.” After carefully rechecking of this link, the link is not further available, thus we replaced with citation number 77 written as “Murray N, Salvatore A, Powell D, Reed-Jones R. Reliability and validity evidence of multiple balance assessments in athletes with a concussion. J Athl Train. 2014;49: 540–549. doi:10.4085/1062-6050-49.3.32.” instead of citation number 76. Moreover, the paper, that may make you wonder about this, is citation number 13 previously written as “Choochouy N, Saita S, Sirithian D. Prevalence of and factors associated with occupational health problems among hill farmers in Thailand. 2022”. Since rechecking we have edited it to correctly read as “Choochouy N, Saita S, Sirithian D. Prevalence of and factors associated with occupational health problems among hill farmers in Thailand. Southeast Asian J Trop Med Public Health. 2022;53: 368–386.” The Southeast Asian Journal of Tropical Medicine and Public Health is in SCOPUS in Q4 as shown in the following link https://www.scimagojr.com/journalsearch.php?q=23093&tip=sid&clean=0.

Section: Methods, Page: 12, Line 280

 Reference, Page: 23, Line: 508-510

Reviewers' comments:

Reviewer's Responses to Questions

Comments to the Author

1. Is the manuscript technically sound, and do the data support the conclusions?

Reviewer #1: Yes

Reviewer #2: Partly

Response: Thank you, we have aimed to clarify. This study began with the cross-cultural adaptation of the BESS instrument from the original English to the Thai language, followed by testing its content and convergent validity, and finally assessing its reliability. 

This study was conducted rigorously. Prior to commencing the study, we reviewed the psychometric properties that the BESS instrument should meet. Additionally, we reviewed previous research and referenced processes most suitable for our study (Chatprem et al., 2020; Georgieva-Zhostova et al., 2014; Jaikaew & Satiansukpong, 2019; Kurre et al., 2009; Meeapirak et al., 2021; Tongprasert et al., 2014), providing detailed descriptions with some pictures for clarity to readers and for replication in future research. 

From statisticians, we obtained confirmation regarding the study sample size, with supporting references. We made efforts to control all potential confounding factors and biases and have rewritten the conclusion more appropriately based on the data support as outlined in the manuscript's relevant sections. 

References

1. Kurre A, Van Gool CJAW, Bastiaenen CHG, Gloor-Juzi T, Straumann D, De Bruin ED. Translation, cross-cultural adaptation and reliability of the german version of the dizziness handicap inventory. Otol Neurotol. 2009;30: 359–367. doi:10.1097/MAO.0B013E3181977E09

2. Tongprasert S, Rapipong J, Buntragulpoontawee M. The cross-cultural adaptation of the DASH questionnaire in Thai (DASH-TH). Journal of Hand Therapy. 2014;27: 49–54. doi:10.1016/J.JHT.2013.08.020

3. Georgieva-Zhostova S, Kolev OI, Stambolieva K. Translation, cross-cultural adaptation and validation of the Bulgarian version of the Dizziness Handicap Inventory. Qual Life Res. 2014;23: 2103–2107. doi:10.1007/S11136-014-0660-5

4. Jaikaew R, Satiansukpong N. Movement assessment battery for children-second edition (MABC2): Cross-cultural validity, content validity, and interrater reliability in Thai children. Occup Ther Int. 2019;2019. doi:10.1155/2019/4086594

5. Meeapirak P, Hunsawong T, Chatchawan U, Siritaratiwat W, Boonprakob Y. Translation, cross-cultural adaptation, and psychometric properties of the Thai version of the Identification of Functional Ankle Instability (IdFAI-THAI) questionnaire. Foot Ankle Surg. 2021 [cited 20 Apr 2022]. doi:10.1016/J.FAS.2021.10.007

6. Chatprem T, Puntumetakul R, Boucaut R, Wanpen S, Chatchawan U. A Screening Tool for Patients With Lumbar Instability: A Criteria-related Validity of Thai Version. Spine (Phila Pa 1976). 2020;45: E1431–E1438. doi:10.1097/BRS.0000000000003606

2. Has the statistical analysis been performed appropriately and rigorously?

Reviewer #1: Yes

Reviewer #2: I Don't Know

Response: Thank you for your review. We have strengthened our reporting of how we analyzed our data appropriately and rigorously as described follows.

For the cross-cultural translation and development of the BESS Thai version phase, the researchers followed and adapted the standard protocol that was reported in previous research (Beaton et al., 2000; Sousa & Rojjanasrirat, 2011) to describe the detail and sequence of the translation phase.

To assess content validity, researchers can use one of the following statistical tests: the Index of Item Objective Congruence (IOC), the Content Validation Index (CVI), or the Content Validity Ratio (CVR). IOC can be used to calculate either multidimensional or unidimensional indexes (Turner & Carlson, 2003). In the current study, the investigators decided to use the IOC because it aims to determine the utility of the BESS-TH in measuring balance among participants with chronic non-specific neck pain, as rated by experts in the relevant context.

Assessing convergent validity, involves determining the degree of correlation between the newly developed scale and other variables, as well as other measures assessing the same construct (Krabbe, 2017). The current study investigators decided to use convergent validity because other tools used to measure static balance in participants with neck pain include: force plates (Jørgensen et al., 2011; Palmgren et al., 2009; Poole et al., 2008; Sremakaew et al., 2021), Single Leg Balance Test (SLBT) (Duray et al., 2018), Romberg test (Jørgensen et al., 2011), Tandem stance test (Treleaven, Jull, et al., 2005; Treleaven, Murison, et al., 2005), and balance error scoring system (BESS) (Wah et al., 2021). 

When comparing the BESS with measures using a force plate (considered the gold standard), it is called “criterion-related validity” as this has already been conducted in previous research and reported (Kleffelgaard et al., 2018). In the current study, we included the clinical field tests of balance measurement in participants with neck pain including the: Single Leg Balance Test (SLBT), Romberg test, Tandem stance test, and balance error scoring system (BESS). 

For assessing reliability in this study, the investigators measured inter and intra-rater reliability. Inter-rater and intra-rater reliability were calculated by the Interclass Correlation Coefficient (ICC). ICC model 2, 1 was used for inter-rater reliability, and model 3, 1 was used for intra-rater reliability. 

However, we have considered the reviewer’s comment, and have added the ceiling and flooring effect, standard error of measurement (SEM), and minimum detectable change (MDC) in our manuscript to cover the objectives of our study.

Section: Abstract, Page: 2, Line: 44-45; 

 Methods, Page: 10, Line: 238-242 and Page: 13, Line: 300-301; 

 Results, Page: 17-18, Line: 361-369 and table 6; 

 Discussion, Page: 19, Line: 380-381, Page: 20, table 7, Page 21, Line: 427-430

References

1. Sousa VD, Rojjanasrirat W. Translation, adaptation and validation of instruments or scales for use in cross-cultural health care research: A clear and user-friendly guideline. J Eval Clin Pract. 2011;17: 268–274. doi:10.1111/j.1365-2753.2010.01434.x

2. Beaton DE, Bombardier C, Guillemin F, Ferraz MB. Guidelines for the process of cross-cultural adaptation of self-report measures. Spine (Phila Pa 1976). 2000;25: 3186–3191. doi:10.1097/00007632-200012150-00014

3. Turner RC, Carlson L. Indexes of Item-Objective Congruence for Multidimensional Items. Int J Test. 2003;3: 163–171. doi:10.1207/s15327574ijt0302_5

4. Krabbe PFM. Validity. The Measurement of Health and Health Status. 2017; 113–134. doi:10.1016/B978-0-12-801504-9.00007-6

5. Poole E, Treleaven J, Jull G. The influence of neck pain on balance and gait parameters in community-dwelling elders. Man Ther. 2008;13: 317–324. doi:10.1016/j.math.2007.02.002

6. Jørgensen MB, Skotte JH, Holtermann A, Sjøgaard G, Petersen NC, Søgaard K. Neck pain and postural balance among workers with high postural demands - A cross-sectional study. BMC Musculoskelet Disord. 2011;12. doi:10.1186/1471-2474-12-176

7. Sremakaew M, Treleaven J, Jull G, Vongvaivanichakul P, Uthaikhup S. Altered neuromuscular activity and postural stability during standing balance tasks in persons with non-specific neck pain. Journal of Electromyography and Kinesiology. 2021;61: 102608. doi:10.1016/j.jelekin.2021.102608

8. Palmgren PJ, Andreasson D, Eriksson M, Hägglund A. Cervicocephalic kinesthetic sensibility and postural balance in patients with nontraumatic chronic neck pain - A pilot study. Chiropr Osteopat. 2009;17: 1–10. doi:10.1186/1746-1340-17-6

9. Duray M, Simşek S, Altuğ F, Cavlak U. Effect of proprioceptive training on balance in patients with chronic neck pain. Agri. 2018;30: 130–137. doi:10.5505/agri.2018.61214

10. Treleaven J, Murison R, Jull G, LowChoy N, Brauer S. Is the method of signal analysis and test selection important for measuring standing balance in subjects with persistent whiplash? Gait Posture. 2005;21: 395–402. doi:10.1016/J.GAITPOST.2004.04.008

11. Treleaven J, Jull G, LowChoy N. Standing balance in persistent whiplash: A comparison between subjects with and without dizziness. J Rehabil Med. 2005;37: 224–229. doi:10.1080/16501970510027989

12. Wah SW, Puntumetakul R, Boucaut R. Effects of proprioceptive and craniocervical flexor training on static balance in university student smartphone users with balance impairment: A randomized controlled trial. J Pain Res. 2021;14: 1935–1947. doi:10.2147/JPR.S312202

13. Kleffelgaard I, Langhammer B, Sandhaug M, Pripp AH, Søberg HL. Measurement properties of the modified and total Balance ERror SCoring System–the BESS, in a healthy adult sample. Eur J Physiother. 2018;20: 25–31. doi:10.1080/21679169.2017.1352020

3. Have the authors made all data underlying the findings in their manuscript fully available?

The PLOS Data policy requires authors to make all data underlying the findings described in their manuscript fully available without restriction, with rare exceptions (please refer to the Data Availability Statement in the manuscript PDF file). The data should be provided as part of the manuscript or its supporting information or deposited in a public repository. For example, in addition to summary statistics, the data points behind means, medians, and variance measures should be available. If there are restrictions on publicly sharing data—e.g. participant privacy or use of data from a third party—those must be specified.

Reviewer #1: Yes

Reviewer #2: Yes

Response: Thank you. The authors plan to add the raw data and the BESS Thai version in the supporting information section as follows. 

“Supporting information.

S1 Data. Analysis Data.

 (XLSX)

S2 Text. A Thai version of the Balance Error Scoring System.

 (PDF)”

Section: Supporting information; Page 22; Line 460-464

4. Is the manuscript presented in an intelligible fashion and written in standard English?

Reviewer #1: Yes

Reviewer #2: Yes

Response: Thank you. 

5. Review Comments to the Author

Reviewer #1: Thank you for the opportunity to review this manuscript, "Validity and reliability of the Balance Error Score System (BESS) Thai version in patients with chronic non-specific neck pain". The manuscript was well written and is appropriate for publication in PLOS ONE Journal.

Response: Thank you for your review and your appreciation of this manuscript.

Reviewer #2: Dear Authors

Thanks for your interesting article. It is very useful in clinical work as well as research.

I have some suggestions as follows:

1. How did you diagnose the patients to have non-specific neck pain? I mean how did you assure that these patients did not have disc herniation or so on?

Response 1: Thank you. Non-specific neck pain (NSNP) is defined as a type of neck pain without a detectable etiology and with no features of red flag conditions such as: malignancy, infection, inflammation, myelopathy, other histories of orthopedics conditions and drop attacks during head movement, or symptoms following whiplash (Binder, 2007; Borghouts et al., 1998). We have added further information in the Introduction. Even though we did not include the patients who have red flag signs, there may be other conditions. For an accurate diagnosis, clinical images such as X-ray, CT, or MRI confirm the true NSNP patients (Parikh et al., 2019). The current study did not provide the opportunity for such diagnostic tests (which are expensive and may be unnecessary). Thus, we put this in the Limitation of the study as follows.

“NSNP is defined as a type of NP without a detectable etiology and with no features of red flag conditions such as malignancy, infection, inflammation, myelopathy, other histories of orthopedics conditions and drop attacks during head movement, or symptoms following whiplash”

“Patients with CNSNP who lacked an identifiable cause and did not exhibit any symptoms of serious underlying illnesses.”

“This study included participants with chronic non-specific neck pain (CNSNP) who met the criteria of neck pain without an identifiable cause and had no indications of serious medical conditions such as cancer, infection, inflammation, myelopathy, previous orthopedic conditions, drop attack during head movement, or symptoms following whiplash injury. Our focus was solely on medical histories, meaning there may have an opportunity for a person with mild cervical disc herniation to have been included in the study. In future studies, investigators may consider the possibility of employing imaging techniques to rule out disc herniation.” 

Section: Introduction, Page 3, Line: 58-61 

 Methods, Page: 6, Line: 144-145

 Discussion, Page 21; Line 436-443

References 

1. Borghouts JAJ, Koes BW, Bouter LM. The clinical course and prognostic factors of non-specific neck pain: a systematic review. Pain. 1998;77: 1–13. doi:10.1016/S0304-3959(98)00058-X

2. Binder A. The diagnosis and treatment of nonspecific neck pain and whiplash. Eura Medicophys. 2007;43: 79–89. 

3. Parikh P, Santaguida P, Macdermid J, Gross A, Eshtiaghi A. Comparison of CPG’s for the diagnosis, prognosis and management of non-specific neck pain: a systematic review. BMC Musculoskelet Disord. 2019;20: 81. doi:10.1186/s12891-019-2441-3

2. Please mention about the “Floor or ceiling effects” in the article. It should be calculated.

Response 2: Thank you. Although neither the “Floor nor ceiling effects” are one of our objectives, they reflect possible reasons to support the correlation between the tests in the convergent validity process. Therefore, we have added the floor and ceiling effects in the Discussion Section and Table 7.

Section: Discussion, Page 20, Table 7

3. In addition to ICC, which has reported in the manuscript, please add the Cronbach’s alpha as well as minimally metrically detectable change (MMDC) for reliability and internal consistency.

Response 3: Thank you for your suggestion. Regarding Cronbach’s alpha, as Reviewer 2 mentioned, it is mostly employed to evaluate the internal consistency of a questionnaire or survey comprising many sub-scales and items (Tavakol & Dennick, 2011). However, the BESS-TH is an instrument that assesses static balance ability, which may differ from other questionnaires such as RMDQ and SF-36, etc. The BESS-TH test has 6 subtests, each measuring different positions and supporting surfaces. Therefore, we have decided not to include Cronbach’s alpha as one of our outcomes.

Regarding the minimally detectable change (MDC) that Reviewer 2 mentioned, the concept of minimal detectable change (MDC) has been used to define the amount of change in a variable that must be achieved before we can be confident that error does not account for the entire measured difference, and that some true change must have occurred (Portney, 2020). In this part, we have decided to add the measurement of MDC and standard error of measurement (SEM) in the reliability phase. Thus, we added the details in the following section. 

Section: Abstract: Page 2; Line 44-45 

 Methods: Page: 10, Line: 238-242 and Page: 13, Line: 300-301

 Results: Page: 17-18, Line: 361-369 and Table 6

 Discussion: Page: 19, Line: 380-381 and Page: 21, Line: 427-430

References

1. Tavakol M, Dennick R. Making sense of Cronbach’s alpha. International journal of medical education. 2011. pp. 53–55. doi:10.5116/ijme.4dfb.8dfd

2. Portney LG. Foundations of clinical research : applications to evidence-based practice. 4th Edition. Philadelphia, PA: F. A. Davis Company; 2020. 

4. Please check the spell of the word “Kolmogorov–Smirnov test”.

Response 4: Kolmogorov–Smirnov is the correct wording for the test – we have checked this in the manuscript.

Section: Methods: Page: 13, Line: 297

5. As you mentioned, you have not checked the criterion validity or using a gold standard to check the score. So, it seems that this article is somehow a cross-cultural adaptation. So, why did not mention in the topic?

Response 5: Thank you for your question. The current study aimed to translate the BESS balance test into the Thai language and perform content and convergent validity, and reliability. In the title, we used the phrase "the Balance Error Score System (BESS) Thai version", indicating the necessity of a cross-cultural adaptation process to obtain this Thai version. Additionally, in the topic, the research employed the term "validity" and provided detailed descriptions of measuring content and convergent validity.

Regarding comparison with force plate measures (considered the gold standard), this was studied in male athletes with correlations ranging from low to high (r=0.31-0.79) (Riemann et al., 1999). This already establishes criterion-related validity, and the current study also compares the clinical field tests that did not include the force plate. Thus, for further research, the researchers mentioned another validity type that was not performed at this time, which is "criterion-related validity" in participants with chronic non-specific neck pain.

Reference

1. Riemann BL, Guskiewicz KM, Shields EW. Relationship between clinical and forceplate measures of postural stability. J Sport Rehabil. 1999;8: 71–82. doi:10.1123/jsr.8.2.71

6. What about the “Dimensionality”? Have you checked it?

Response 6: Thank you for your question. 

 “Unidimensionality” was explained in the book of Portney (2020) as the one the way to examine the internal construct validity as follows. 

“The Rasch model has provided a way to examine the internal construct validity of a measure, including ordering of categories, unidimensionality, and whether items are biased across subgroups (Portney, 2020).” 

The dimensionality of a tool is assessed using factor analysis. Factor analysis aims to identify common patterns among a group of variables by examining their shared variance. The primary goal of this analysis is to determine the most straightforward and concise method for interpreting and displaying the collected data (Manzar et al., 2018). From the review, the studies that perform dimensionality are mainly in questionnaires (Manzar et al., 2018; Rawang et al., 2020; Terwee et al., 2007; Vet et al., 2005). The current study consists of 6 subtests of BESS-TH and 3 other balance tests. Thus, we have considered not analyzing dimensionality.

References

1. Portney LG. Foundations of clinical research : applications to evidence-based practice. 4th Edition. Philadelphia, PA: F. A. Davis Company; 2020. 

2. Manzar MD, BaHammam AS, Hameed UA, Spence DW, Pandi-Perumal SR, Moscovitch A, et al. Dimensionality of the Pittsburgh Sleep Quality Index: a systematic review. Health Qual Life Outcomes. 2018;16. doi:10.1186/S12955-018-0915-X

3. Vet HCW de, Adèr HJ, Terwee CB, Pouwer F. Are factor analytical techniques used appropriately in the validation of health status questionnaires? A systematic review on the quality of factor analysis of the SF-36. Quality of Life Research. 2005;14: 1203–1218. doi:10.1007/s11136-004-5742-3

4. Rawang P, Janwantanakul P, Correia H, Jensen MP, Kanlayanaphotporn R. Cross-cultural adaptation, reliability, and construct validity of the Thai version of the Patient-Reported Outcomes Measurement Information System-29 in individuals with chronic low back pain. Quality of Life Research. 2020;29: 793–803. doi:10.1007/s11136-019-02363-x

5. Terwee CB, Bot SDM, de Boer MR, van der Windt DAWM, Knol DL, Dekker J, et al. Quality criteria were proposed for measurement properties of health status questionnaires. J Clin Epidemiol. 2007;60: 34–42. doi:10.1016/j.jclinepi.2006.03.012

---

## [Decision Letter · Decision Letter 1]

14 Mar 2024

Validity and reliability of the Balance Error Score System (BESS) Thai version in patients with chronic non-specific neck pain

PONE-D-23-41694R1

Dear Dr. Thiwaphon Chatprem

We’re pleased to inform you that your manuscript has been judged scientifically suitable for publication and will be formally accepted for publication once it meets all outstanding technical requirements.

Kind regards,

Mehrnaz Kajbafvala, Ph.D

Academic Editor

PLOS ONE

Additional Editor Comments (optional):

Reviewers' comments:

Reviewer's Responses to Questions

**Comments to the Author**

1. If the authors have adequately addressed your comments raised in a previous round of review and you feel that this manuscript is now acceptable for publication, you may indicate that here to bypass the “Comments to the Author” section, enter your conflict of interest statement in the “Confidential to Editor” section, and submit your "Accept" recommendation.

Reviewer #2: All comments have been addressed

Reviewer #3: All comments have been addressed

2. Is the manuscript technically sound, and do the data support the conclusions?

Reviewer #2: Yes

Reviewer #3: Yes

3. Has the statistical analysis been performed appropriately and rigorously? 

Reviewer #2: I Don't Know

Reviewer #3: Yes

4. Have the authors made all data underlying the findings in their manuscript fully available?

Reviewer #2: Yes

Reviewer #3: Yes

5. Is the manuscript presented in an intelligible fashion and written in standard English?

Reviewer #2: Yes

Reviewer #3: Yes

6. Review Comments to the Author

Reviewer #2: (No Response)

Reviewer #3: In my opinion, the manuscript in its present form is acceptable for publication. I hope the best for the authors.

7. PLOS authors have the option to publish the peer review history of their article (what does this mean?). If published, this will include your full peer review and any attached files.

Reviewer #2: No

Reviewer #3: **Yes: **Mehrnaz Kajbafvala

---

## [Editor Report · Acceptance letter]

19 Mar 2024

PONE-D-23-41694R1 

PLOS ONE

Dear Dr. Chatprem, 

I'm pleased to inform you that your manuscript has been deemed suitable for publication in PLOS ONE. Congratulations! Your manuscript is now being handed over to our production team.

Kind regards, 

on behalf of

Dr. Mehrnaz Kajbafvala 

Academic Editor

PLOS ONE